# Effect of Supplementary Feeding on Milk Volume, Milk Composition, Blood Biochemical Index, and Fecal Microflora Diversity in Grazing *Yili* Mares

**DOI:** 10.3390/ani13152415

**Published:** 2023-07-26

**Authors:** Hao Lu, Wenjie Zhang, Shuo Sun, Yingying Mei, Guodong Zhao, Kailun Yang

**Affiliations:** Xinjiang Key Laboratory of Meat & Milk Production Herbivore Nutrition, College of Animal Science and Technology, Xinjiang Agricultural University, Urumqi 830052, China; seven_lh_eleven@163.com (H.L.); xndzhangwenjie@163.com (W.Z.); s1161931970@163.com (S.S.); m15628268756@163.com (Y.M.)

**Keywords:** fecal flora, milk composition, supplementary feeding, *Yili* horse

## Abstract

**Simple Summary:**

Grazing mares were fed concentrate and coated fatty acids via a customized pocket. The milk yield and composition, plasma biochemistry, and fecal flora were compared among Yili mares that either grazed normally or were fed dietary supplements. Milk yield and fat, lactose, and protein levels were significantly higher among the horses fed dietary supplements compared to those that grazed. All the supplementary diets improved the horses’ milk fatty acid profiles, while coated fatty acid supplementation increased plasma glucose levels. Supplementary feeding significantly increased the fecal abundance of Verrucomicrobia, which regulate lipid biosynthesis, metabolism, and energy metabolism.

**Abstract:**

Grazing is a common approach to rearing. We investigated the effects of supplementation during grazing on milk yield and composition, blood biochemistry, and fecal microflora in *Yili* horses. The control mares grazed normally, while those in groups I and II received 1 kg/d of concentrate and 1 kg/d of concentrate + 0.4 kg/d of coated FA, respectively. Milk volumes were significantly higher in groups I and II than in the control group, and among the previous two, milk volumes were significantly higher in group II than in group I. Milk fat, lactose, and protein levels were significantly higher in group II than in the others. BUN was highly significantly lower in group I than in the control group. Specific FAs, total SFA, and total UFA were significantly higher in group II than in the other groups. After feeding, plasma GLU, free FA, TG, LDL, and VLDL were significantly higher in group II than in the other groups. The control group, group I, and group II had 4984, 5487, and 5158 OTUs, respectively, and 3483 OTUs were common to all groups. The abundance of Bacteroidetes and Firmicutes was >75%. The abundance of Verrucomicrobia was significantly higher in groups I and II than in the control group and, among the previous two, significantly higher in group II than in group I. The abundance of *Treponema_saccharophilum* significantly differed between the control and other groups, and *WCHB 1_41*, Kiritimatiellae, and Verrucomicrobia abundances significantly differed between groups II and the other groups.

## 1. Introduction

The *Yili* dairy horse is independently bred in Xinjiang, China. It has excellent lactation performance, adapts well to a wide range of environmental conditions, and tolerates severe cold and short pasture growth periods. Milk yield and quality determine the normal growth and development of the offspring [1]. This milk also has high economic value as its nutritional composition and digestibility are similar to those of human milk [2]. Due to the low production levels, high cost, and storage difficulties relating to horse milk, it is expensive and cannot be industrially mass-produced. Moreover, in practice, the *Yili* horse’s milk yield is low during the animal’s 5–8-month lactation period and cannot simultaneously accommodate both human and livestock demands [3]. Thus, the milk produced does not meet the growth, development, and nutritional requirements of the foals. Furthermore, nutrient and energy requirements substantially increase during lactation. In fact, the energy demand rises by 44% during milk production compared to other life cycle periods [4]. Grazing alone does not ensure optimal milk yields and quality. Hence, increasing these parameters or adding nutrient supplements to grazing is necessary to enhance and maintain the healthy growth and development of foals and the performance of mares. Fats have a higher energy content and density than either proteins or carbohydrates [5]. At the same time, increasing the output of horse milk can increase the income of herdsmen, thus promoting enthusiasm among herdsmen with regard to raising horses and promoting the sustainable and healthy development of the horse milk industry. Grazing and supplementary feeding can also reduce the increased risk of certain diseases caused by the high-density breeding of horses [6,7,8]. Raising mares using the grazing approach increases their free feeding ability and reduces the incidence of diseases, such as stomatitis, caused by the long-term consumption of dry grass [9]. Dietary fat improves feed palatability [10], supplies essential fatty acids (FAs), facilitates fat-soluble vitamin absorption and utilization, and reduces feed dust [11]. Spers et al. (2006) found that dietary oil supplementation significantly improved lactation performance among mares. Among mares maintained on a high-energy diet, Doreau et al. (1992) reported that the mares’ milk yield increased by ~10% but that their milk fat and protein content slightly declined [12]. Exogenous FA supplementation enables the mammary glands to absorb long-chain and certain C16 FAs from the blood and synthesize triglycerides that are polymerized into milk fat droplets and secreted into the milk [13]. In this way, the fat and FA levels are increased in horse milk [14]. Milk quality, intestinal immunity, and energy supply are closely associated with the gut microbiome among horses [15]. Exogenous supplementary feeding might help maintain the gut microflora in horses [16]. The present study aimed to explore the effects of supplementary feeding on milk yield and composition and fecal microflora diversity among grazing *Yili* horses and provide a reference for improving the yield and quality of the milk and, by extension, favor the healthy growth and development of foals.

## 2. Materials and Methods

### 2.1. Trial Date and Location

The present study was conducted in Zhaosu *Yili* Kazak Autonomous Prefecture (E 80°83′23″, N 42°83′12″) over a 63 d period between August and October 2022. All procedures in this study were approved by the Animal Experiment Ethics Committee of Xinjiang Agricultural University (permit number: 2018012).

### 2.2. Experimental Design

Eighteen healthy *Yili* mares that had similar parameters in terms of foal birth date, lactation, milk yield, and physiological state were purchased from Xinjiang *Yili* Prefecture Zhaosu County Junli Agriculture and Animal Husbandry Technology Co., Ltd. (Xinjiang, China) and randomly assigned to a control group or test groups I or II. Each group had six horses. The animals in the control group fed via grazing alone, while those in group I were grazed and received 1 kg/d of feed concentrate and those in group II grazed and received 1 kg/d of feed concentrate plus 0.4 kg/d of coated FAs.

### 2.3. Feed Management

All horses grazed in the same pasture. Supplementary feeding started at 10:30 every day, during which the mares were separated from their foals. The animals in the control group were fitted with an empty customized pocket, while those in groups I and II wore full pockets containing feed concentrate or feed concentrate + fat powder, respectively. After the horses in experimental groups I and II had finished eating the concentrate, the customized pockets were removed from all the horses. Milk was first collected at 11h00 daily and once every 2 h thereafter. After the final milk collection procedure at 19h00, the foals and mares were driven to their designated pastures for grazing. After 3 d of pre-feeding, the experiment was initiated. All horses were managed in the same way and had *ad libitum* food and water access. The nutrient compositions of the feed concentrate supplement are listed in Table 1. The FA compositions of the feed concentrate, pasture, and fat powder are listed in Table 2.

### 2.4. Sample Collection and Analysis

#### 2.4.1. Milk

Samples were collected via manual milking every 7 d. When milking, the mares were taken to the enclosure of the foals, and then the foals were led to the mares to allow them to suck the mares’ nipples for 2–5 s to stimulate the mammary gland to secrete milk. The foals and the mares were then separated, and the milking staff began to collect the mares’ milk, keeping the foals beside the mares to calm them, with the aim of collecting more horse’s milk. Ten percent of each daily milk yield was placed in 100 mL milk sample bottles and stored at −20 °C.

#### 2.4.2. Blood

On day 40, 5 mL of blood was drawn from the jugular vein at 0, 1, 2, 3, 4, 6, 9, and 12 h after the morning feeding; placed in a centrifuge tube containing heparin sodium; and centrifuged at 4000× *g* for 15 min. The plasma was transferred to 2.0 mL Eppendorf tubes (Eppendorf GmbH, Hamburg, Germany) and stored at −20 °C.

#### 2.4.3. Feces

On days 45 and 46, feces were collected between 11:30 and 19:30 and pooled within 8 h. After 2 d, the samples were mixed, and 10% of the homogenate was weighed, air-dried, ground to particles 0.63 mm in diameter, and stored. A total of 5 grams of fresh feces was placed in each 5 mL storage tube and frozen in liquid nitrogen.

### 2.5. Sample Analysis

#### 2.5.1. Daily Lactation Volume

At each of the five daily milkings, the milk volume was measured, the milk was weighed on an electronic scale, the total daily milk volume was calculated, and all values were recorded.

#### 2.5.2. Milk Composition

On days 21, 28, 42, and 49, frozen milk samples were thawed via preheating in a water bath at 42 °C, cooled to 20–25 °C, slowly homogenized through ~20 serial inversions, and placed in an automatic milk composition analyzer (No. UL80BC-11; Hangzhou Zhejiang University Optimal Technology Co., Ltd., Hangzhou, China). Certain milk samples were sent to the Shandong Academy of Agricultural Sciences (Jinan, Shandong, China), where their FA content was evaluated using high-efficiency gas chromatography (GC) [17].

#### 2.5.3. Plasma Biochemical Indices

On day 40, plasma was collected at 0, 1, 2, 3, 4, 6, 9, and 12 h after the morning feeding and sent to the Beijing Huaying Institute of Biotechnology, where the glucose, free fatty acid (FFA), triglyceride, low-density lipoprotein (LDL), and very-low-density lipoprotein (VLDL) levels of these plasma samples were measured via colorimetry [18].

#### 2.5.4. Fecal Microflora Abundance and Diversity

Fecal samples were sent to Beijing Nuohe Zhiyuan BioInformation Technology Co., Ltd. (Beijing, China), where the 16S rDNA V3-V4 region of the intestinal bacteria was selected, diluted, and used for PCR amplification with specific primers. The resultant cDNA was double-end-spliced and subjected to quality control (QC), and chimeras were filtered out. Libraries were sequenced from the purified DNA on the Illumina HiSeq platform (Illumina, San Diego, CA, USA) and used to distinguish the bacterial species compositions and community structures among the samples [19,20,21,22].

### 2.6. Data Analyses

First, all data were sorted in Microsoft Excel 2010 (Microsoft Corp., Redmond, WA, USA). Milk volumes and plasma biochemical indices were analyzed using GraphPad Prism v. 8.0.2 (GraphPad Software, La Jolla, CA, USA). Milk composition was analyzed via two-way ANOVA in SAS v. 8.1 (SAS Institute, Cary, NC, USA). Fecal microflora structure and diversity were analyzed via ANOVA in SPSS v. 21.0 (IBM Corp., Armonk, NY, USA). Duncan’s multiple-range test method was used for multiple comparisons. Significant trends, significant differences, and very significant differences were designated at 0.05 ≤ *p* < 0.10, *p* < 0.05, and *p* < 0.01, respectively.

## 3. Results

### 3.1. Effects of Concentrate and FA Supplementation on Milk Volume in Grazing Yili Horses

The daily average and total milk volumes peaked in the first week and gradually decreased thereafter (Figure 1). The daily average and total milk volumes of group II were significantly higher than those of the control group (31.72% and 21.28%) and group I (31.73% and 21.33%) (*p* < 0.05). The daily average and total milk volumes of group I were significantly higher than those of the control group (8.61% and 8.57%) (*p* < 0.05) (Figure 2).

### 3.2. Effects of Feed Concentrate and FA Supplementation on the Composition of Grazing Yili Horse Milk

Table 3 shows that there were no significant differences between the groups in terms of milk fat, milk protein, lactose, and total solids content or somatic cell number (*p* > 0.05). The milk fat (32.87% and 27.46%), milk protein (33.21% and 23.90%), and lactose levels (18.50% and 21.23%) were significantly higher in group II than in either the control or group I (*p* < 0.05). The urea nitrogen content (9.90% and 8.39%) was significantly lower in groups I and II than in the control (*p* < 0.05) and very significantly decreased in a time-dependent manner (*p* < 0.01).

### 3.3. Effects of Feed Concentrate and FA Supplementation on FA Composition of Grazing Yili Horse Milk

Table 4 shows that the concentrations of lauric (123.41% and 46.20%), tetradecanoic (20% and 22.62%), and *cis*-11,14,17-eicosatrienoic acids (41.38% and 103.24%) and those of the total saturated fatty acids (SFAs) (32.98% and 38.46%) and unsaturated fatty acids (UFAs) (14.43% and 26.15%) were significantly higher in group II than in the control group or group I (*p <* 0.05). The concentrations of heptadecanoic (32.61% and 31.49%), monoenoic (34.14% and 16.19%), γ-linolenic (26.58% and 28.26%), α-linolenic (25% and 22.32%), total linolenic (11% and 18.41%), and total polyunsaturated fatty acids (PUFAs) were significantly lower in group II than in the control group or group I (*p* < 0.05). The concentrations of pentadecanoic (39.13%) and linoleic acids (9.12%) were significantly lower in group II than in the control group (*p* < 0.05).

### 3.4. Effects of Feed Concentrate and FA Supplementation on the Plasma Biochemical Indices of Grazing Yili Horses

For groups I and II, the plasma glucose (GLU) concentration (Figure 3) initially increased, reached its peak at 1 h after feeding, and decreased thereafter. At 1 h (58.97% and 21.63%), 2 h (12.50% and 41.98%), and 9 h (19.94% and 18.42%) after feeding, the plasma GLU concentration was significantly higher in group II than in either the control or group I (*p* < 0.05). At 4 h after feeding, the plasma GLU concentration (16.34%) was significantly higher in group II than in group I (*p* < 0.05).

The plasma FFA (Figure 4) and triglyceride (Figure 5) concentrations were highest at 0 h and lowest at 1 h after feeding. Overall, the plasma FFA and triglyceride concentrations gradually increased after feeding.

At 3 h and 6 h after feeding, the plasma FFA (50.00% and 50.00%; 80.00% and 200.00%) and triglyceride concentrations (100.00% and 38.46%; 47.62% and 93.75%) were significantly higher in group II than in the control group and group I (*p* < 0.05). At 3 h after feeding, the plasma FFA and triglyceride concentrations were significantly higher in group II than in group I (*p* < 0.05). At 2 h after feeding, the plasma FFA concentrations were higher in group II than in group I (0.05 < *p* < 0.10). The plasma low-density lipoprotein (LDL) (Figure 6) and very low-density lipoprotein (VLDL) (Figure 7) concentrations in groups I and II remained relatively stable after feeding. At 3 h and 9 h after feeding, the plasma LDL (27.27% and 40%; 18.92% and 18.92%) and VLDL (40% and 55.56%; 25.71% and 33.33%) concentrations, respectively, were significantly higher in group II than in either the control group or group I (*p* < 0.05).

### 3.5. Effects of Feed Concentrate and FA Supplementation on Fecal Microflora Diversity in Grazing Yili Horses

Flattening the dilution curve indicated that the number of fecal microflora samples sufficed for an alpha-diversity analysis (Figure 8A). There were 4984, 5487, and 5158 OTUs in the control group and groups I and II, respectively. Of these, 3483 OTUs were common to all three groups (Figure 8B). Hence, relatively more bacterial species were detected in groups I and II than in the control group, and concentrate and fatty acid supplementation increased intestinal microflora diversity.

### 3.6. Effects of Feed Concentrate and FA Supplementation on Alpha Diversity of Fecal Bacterial Phyla in Grazing Yili Horses

Table 5 shows that coverage was 0.99 for each group. Thus, the data accurately reflected the horses’ intestinal microflora composition. The Shannon, Chao1, and ACE indices were non-significantly higher in groups I and II than in the control group (*p* > 0.05). Therefore, the feed concentrate and FA supplementation partially improved intestinal microflora abundance and diversity.

### 3.7. Effects of Feed Concentrate and FAs on Fecal Bacterial Phylum Abundance in Grazing Yili Horses

Table 6 shows the composition and relative abundance of the fecal bacterial phyla Bacteroidetes, Firmicutes, Spirochaetes, Verrucomicrobia, Unidentified_Bacterri, Proteobacteria, Euryarchaeota, Halobacterota, Fibrobacterota, and Acidobacteriota. The total abundance of Bacteroidetes and Firmicutes was >75%. The abundance of Verrucomicrobia was significantly higher in groups I (73.72%) and II (102.04%) than in the control group (*p* < 0.05) and significantly higher in group II than in group I (16.30%) (*p* < 0.05). The relative abundances of all other foregoing phyla did not significantly differ among groups (*p* > 0.05).

### 3.8. Effects of Feed Concentrate and FA Supplementation on Fecal Bacterial Family Abundance in Grazing Yili Horses

Table 7 shows that the top ten (most abundant) bacterial families were Rikenellaceae, Lachnospiraceae, Spirochaetaceae, Prevotellaceae, p-251-o5, F082, Bacteroidales_RF16_group, Oscillospiraceae, Clostridiaceae, and Ruminococcaceae. However, their abundances did not significantly differ between the groups (*p* > 0.05).

### 3.9. Effects of Concentrate and Fatty Acid Supplementation on Fecal Bacterial Genus Abundance in Grazing Yili Horses 

Table 8 shows that the top ten (most abundant) bacterial genera were Treponema, Rikenellaceae_RC9_gut_group, Clostridium_sensu_stricto_1, Prevotellaceae_UCG-001, UCG-004, Prevotellaceae_UCG-004, Ruminococcus, Prevotellaceae_UCG-003, Lachnospiraceae_UCG-009, and Fecalibaculum. The abundance of Prevotellaceae_UCG-001 was lower in groups I and II than in the control group (0.05 < *p* < 0.10). In contrast, the abundances of all other bacterial genera were significantly higher in groups I (7.02%) and II (7.76%) than in the control group (*p* < 0.05).

### 3.10. LEfSe Analysis and Tax4Fun Function Prediction of Fecal Bacterial Taxa in Grazing Yili Horses Supplemented with Feed Concentrate and FAs

The taxonomic rank of *Treponema_saccharophilum* in the control group significantly differed from those in groups I and II. The taxonomic ranks of *WCHB1_41*, *Kiritimatiellae*, and *Verrucomicrobia* in group II significantly differed from those in the control group and group I (Figure 9A,B).

### 3.11. PICRUSt Functional Prediction

The fecal microflora in groups I and II were implicated in lipid biosynthesis and metabolism, energy metabolism, and amino acid metabolism (Figure 10). Those in group I alone were implicated in lipid biosynthesis and metabolism (Figure 11) as well as genetic information processing and carbohydrate metabolism (Figure 12).

## 4. Discussion

In female mammals, lactation is associated with substantial energy expenditure. In fact, the energy demand is 1.9-fold higher during lactation than at other life cycle periods [23]. An inadequate energy supply induces fat mobilization, which, in turn, results in reductions in weight and lactation. Low pregnancy rates and early embryo deaths occur among undergrazed lactating mares [24]. An inadequate energy intake results in chronic low milk yields [25,26]. Among sows, fat consumption during late pregnancy and early lactation increases milk yields and milk fat content and improves feed efficiency [27]. Here, the average and total daily milk yields were significantly higher in group II than in group I or the control group. Hence, FA supplementation can increase milk yield among grazing mares. During the final three weeks of the trial, the milk yield in group Ⅰ significantly differed from that in the control group as there was a lack of rainfall, the grass was sparse and low in quality, and lactation was late. Diet and nutrition have a stronger impact on the composition of equine milk compared to that of ruminant milk [28]. Davison et al. (1991) showed that lipid supplementation significantly improved milk composition in the corresponding supplementation group compared to the untreated controls [14]. In the present study, milk composition did not significantly differ between groups. During the September trial, the milk fat, lactose, and milk protein concentrations were significantly higher in group Ⅱ than in the others, possibly because the exogenous FAs participated in hydrolysis reactions, thereby promoting lactation. Urea nitrogen content, crude protein intake, and the protein utilization ratio are strongly correlated [29]. The urea nitrogen level was significantly lower in group 1 than in the other groups. Thus, feed concentrate supplementation improves protein utilization in lactating mares. Milk fat is rich in various FFAs with physiologically important functions. In milk, lauric and myristic acids have the highest efficiencies of conversion to other FAs [30]. Lauric acid has strong antibacterial efficacy, while myristic acid stimulates metabolism [31]. In this study, the lauric and myristic acid concentrations were significantly higher in group Ⅱ than in the other groups. High levels of these FAs strengthen the immune response, improve nutrient assimilation and metabolism, and enhance milk production in lactating mares. Dietary lipid type and content strongly influence milk fat composition. Glasser et al. (2008) reported that dietary vegetable oil infusions significantly increased the UFA content and decreased the SFA content in milk [32]. The proportions of FAs in horse and human milk are similar. In both cases, the relative amounts of PUFAs with high carbon numbers are greater in these types of milk than in those derived from other mammals. For this and other reasons, horse milk has high nutritional value for humans [33]. According to the published data, concerning horse milk, the total solid production is about 110 g/kg, lactose production is about 61 g/kg, total nitrogen production is about 21.4 g/kg, fat production is about 14 g/kg, and ash production is about 4.5 g/kg [34,35,36].The total solid yield of human milk is about 125 g/kg, the lactose yield is about 64.4 g/kg, the total nitrogen production is about 12.5 g/kg, and the fat yield is about 34.6 g/kg. Thus, the ash yield and the total solid and lactose production in horse milk are very close to those of human milk [37,38], making horse milk a good substitute for human milk. Most of the fatty acids in horse milk are unsaturated fatty acids and low molecular fatty acids, with a ratio of 1:3, which is similar to that of human milk (1:2), and the content is 4–5 times that of cow milk, which has a good effect on improving human immunity and preventing diseases [39]. Compared with the fatty acids in bovine milk, the melting point of fatty acids in horse milk is usually 5–11 °C lower, and the water content of horse milk is higher, allowing it to be more easily digested and absorbed by the human body. In terms of milk protein, the average protein content in horse milk is closer to that of human milk [40]. Horse milk is rich in the essential amino acids required by the human body. Most of these proteins are types of whole protein, which contain high quantities of albumin and globulin, mainly immunoglobulin, which has the function of enhancing body resistance and regulating energy metabolism. The average content of horse lactose is basically the same as that of human milk and higher than that of cow’s milk [41], Horse milk is more palatable; however, fermented horse milk is not suitable for young animals. This study found that the absolute values of calcium and phosphorus in horse milk are uncertain, but their average values are still higher than those of cow’s milk and are the same as those of human milk. The average value of the calcium to phosphorus ratio in horse milk is 1.72, while the average values of the calcium to phosphorus ratio in human and cow’s milk are 1.7 and 1.23, respectively [41,42], demonstrating that horse milk is more acceptable to the human body. Compared to the other groups, group II’s total SFA content was significantly higher, while its total UFA content was significantly lower; this result is possibly because of the relative differences in fat supplementation between the groups.

Plasma GLU levels reflect energy intake and fat accumulation. Zeyner et al. (2010) found that the blood sugar content increases with dietary fat intake in horses [43]. Plasma GLU is a metabolic energy source and a precursor for more complex cellular components. It also plays important roles in various physiological processes such as locomotion [44]. In the case of insufficient energy supply to the body, some of the exogenous fatty acids form acetyl-coenzyme A through β-oxidation in the TCA cycle to produce a large amount of energy, and some of the fatty acids are converted into blood sugar through sugar metabolism in the acetyl-coenzyme A cycle in order to provide energy for the body. Fatty acids in milk are partly obtained from food, partly from microbial fermentation, and partly from de novo synthesis. FAs improve GLU tolerance and reduce insulin sensitivity [45]. In this study, the blood GLU concentration significantly increased in group Ⅱ at 1, 2, 4, and 9 h after feeding. Hence, FA supplementation significantly increased the energy levels in the mares. An increase in blood sugar causes plasma FFA to be converted in the liver to triglycerides and adipose tissue. Lipoproteins are fat carriers involved mainly in lipid packaging and transport. Lipoprotein levels reflect the extent of lipid metabolism [46]. Here, the plasma FFA, triglyceride, LDL, and VLDL levels increased at 3 h and 9 h after supplementation. Therefore, FA supplementation increases blood GLU uptake by the liver, while grazing facilitates hepatic triglyceride and cholesterol transport. Both mechanisms promote fat metabolism and improve lactation performance among grazing *Yili* mares.

Animal intestines provide a suitable environment for microbial commensals [47]. In exchange, the gut microbiota regulates animal health and growth [48]. Diet is the main factor affecting the gut microbiome’s structure and diversity in animals [49]. In this study, the Shannon, Chao1, and ACE diversity indices were relatively but non-significantly higher in group II than in the others. Nevertheless, the combination of feed concentrate and FAs improved gut microbiota abundance and diversity. The control group, group Ⅰ, and group Ⅱ had 4894, 5487, and 5187 OTUs, respectively. Consequently, the combination of feed concentrate and FAs increased the number of bacterial species compared to the other treatments. The number and structure of intestinal microflora affect the growth, development, and performance of lactating mares [47]. Bacteroides and Planctomycetes facilitate cellulose and carbohydrate digestion and absorption [50,51]. In the present work, there were no significant differences among the groups in terms of Bacteroides or Planctomycetes abundance despite the relative differences between treatments in terms of the types or amounts of FAs administered. Verrucomicrobia supplies metabolic energy to the host, and its metabolites participate in immunoregulation [52]. Thus, Verrucomicrobia regulates intestinal immunity. Spirochetes are highly pathogenic [53]. In this study, the relative abundances of *Microbacterium verruciformis* and *Spirillum* sp. were significantly higher and lower, respectively, in groups I and II than in the control group. Thus, feed concentrate and FA supplementation may help attenuate inflammation and improve immunity in lactating mares. Su et al. (2020) stated that under natural grazing conditions, the dominant gut microbial families in Mongolian horses were Trichospiridae, Prevotellidae, and others. Similar findings were observed in the present study [54]. Members of Prevotellidae degrade hemicellulose and proteins [55]. Both prevol_ucg_003 and prevol_ucg_001 enhance cellulose digestibility [56,57]. Feed concentrate and FA supplementation might affect cellulose digestibility and utilization in grazing horses as certain FAs are bacteriostatic.

An LEfSe analysis revealed that the relative abundance of *Treponema saccharophila* was higher in the control group than in the other groups. This bacterium promotes cellulose digestion and decomposition in the host. *Treponema saccharophila*, *T. brucei*, and *T. purpura* decompose pectin to acetate via *trans*-elimination. The dominant bacterial species in group II were *WCHB1-c1* (Kiritimatiellae) and *Microbacterium verruciformis*. The gut bacteria have an affinity for the mucus layer on the host intestinal epithelium. The status of the mucus layer is a biomarker of gut health. When the relative abundance of the intestinal microbiota declines, the epithelial mucus layer thins, resulting in diarrhea [58]. Members of Kiritimatiellae have an affinity for the intestinal mucus layer [59]. A high relative abundance of Kiritimatiellae favors intestinal mucus production and, by extension, normal gut function [60]. The relative abundance of *WCHB1-41* is reduced in horses afflicted with diarrhea [61]. However, the relationship between *WCHB1-41* and intestinal inflammation remains to be determined [62]. Dietary fat supplementation partially ameliorates equine diarrhea. *Microbacterium verruciformis* is very abundant in the equine gut. It is immunomodulatory and supplies energy to the host [63]. These findings are consistent with the PICRUSt predictions. Microbial community structure and function vary with feeding type [64]. The functional analysis revealed that the dominant intestinal bacterial taxa in group II were implicated in lipid anabolism, energy metabolism, and amino acid metabolism, while those in group I were involved in lipid biosynthesis and metabolism and those in the control group participated in carbohydrate metabolism.

In this study, based on grazing, it was revealed that feeding mares 1 kg of concentrate +0.4 kg of fat powder supplement 30 min before the first collection of horse milk significantly increased their lactation volumes. However, the mares could not be maintained in a calm state at each milking under the grazing conditions during this experiment. In addition, the lactation volumes of mares vary at different times. As a monogastric herbivore, food stays in their gastrointestinal tract for a short period. Refeeding 30 min before the third collection can be employed to improve the overall lactation volumes of mares.

## 5. Conclusions

Grazing supplementation with feed concentrate and FAs may significantly increase the yields and fat, lactose, protein, and FA content and improve the FA composition in the milk of grazing mares. Supplementation can also significantly increase the plasma GLU, triglyceride, LDL, VLDL, and FFA levels in grazing mares and partially improve the abundance and diversity of their intestinal microflora.

## Figures and Tables

**Figure 1 animals-13-02415-f001:**
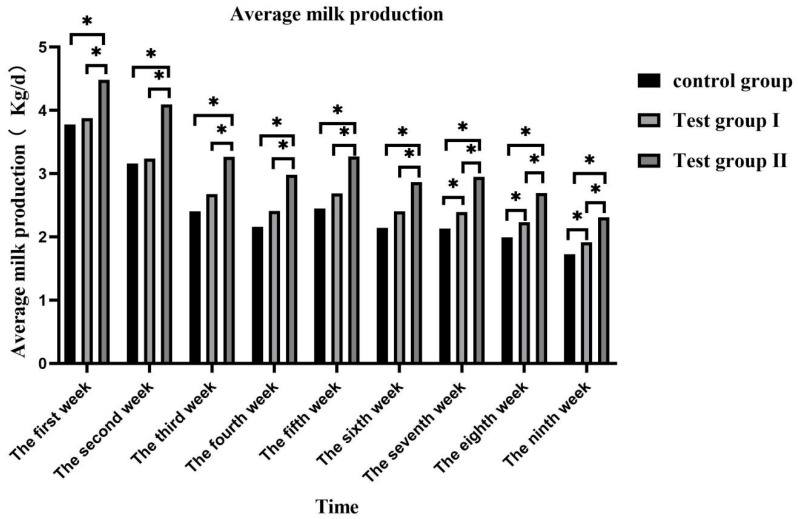
Effects of feed concentrate and FA supplementation on daily average milk production among grazing *Yili* horses. * Significant differences.

**Figure 2 animals-13-02415-f002:**
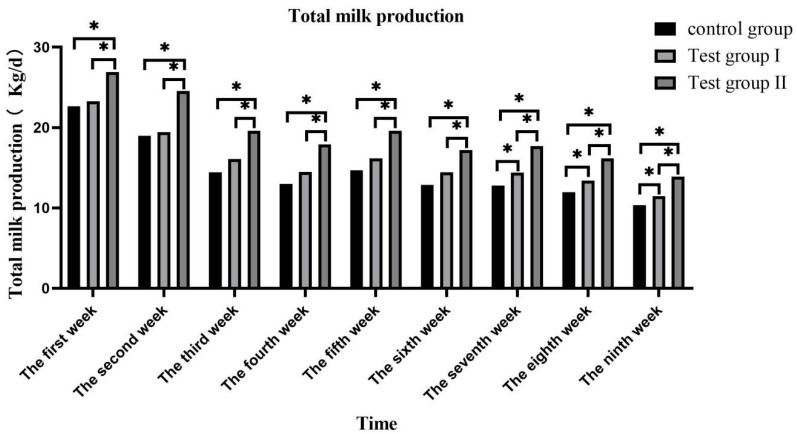
Effects of feed concentrate and FA supplementation on daily total milk production among grazing *Yili* horses. * Significant differences.

**Figure 3 animals-13-02415-f003:**
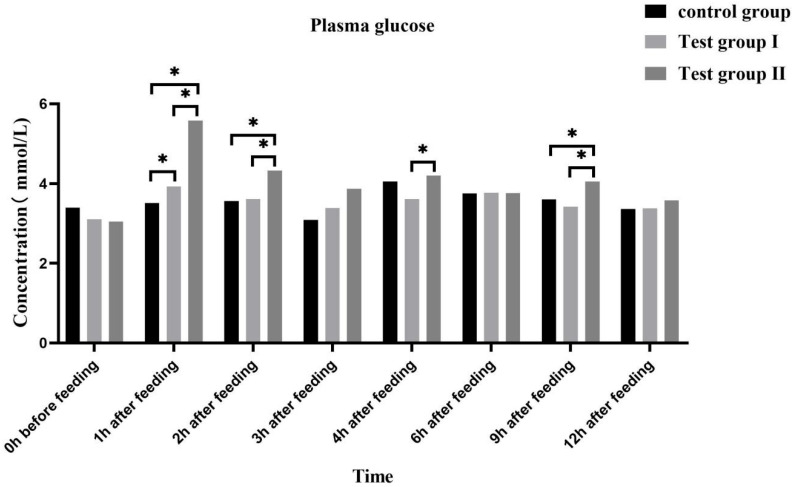
Effects of feed concentrate and FA supplementation on plasma GLU concentrations in grazing *Yili* horses. * Significant differences.

**Figure 4 animals-13-02415-f004:**
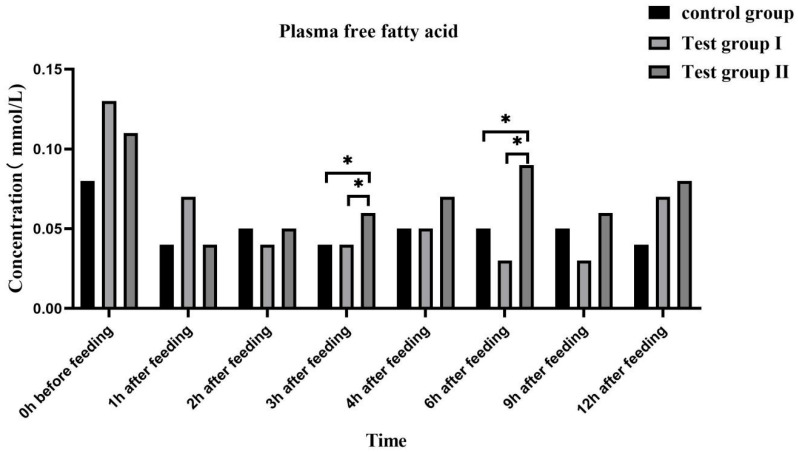
Effects of feed concentrate and FA supplementation on the plasma FFA concentrations in grazing *Yili* horses. * Significant differences.

**Figure 5 animals-13-02415-f005:**
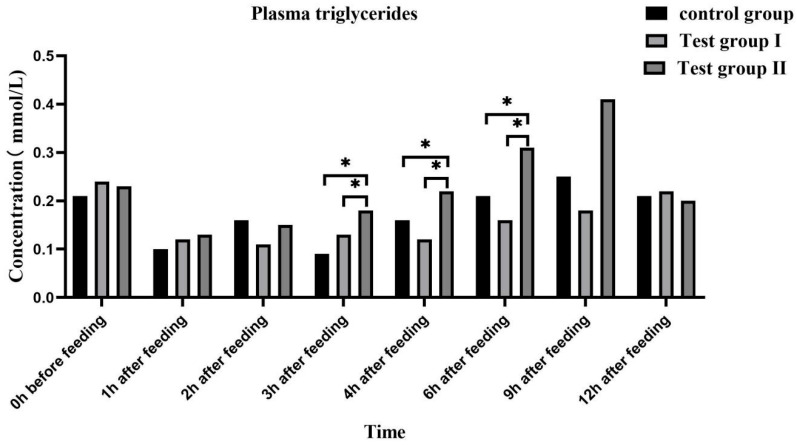
Effects of feed concentrate and FA supplementation on plasma triglyceride concentrations in grazing *Yili* horses. * Significant differences.

**Figure 6 animals-13-02415-f006:**
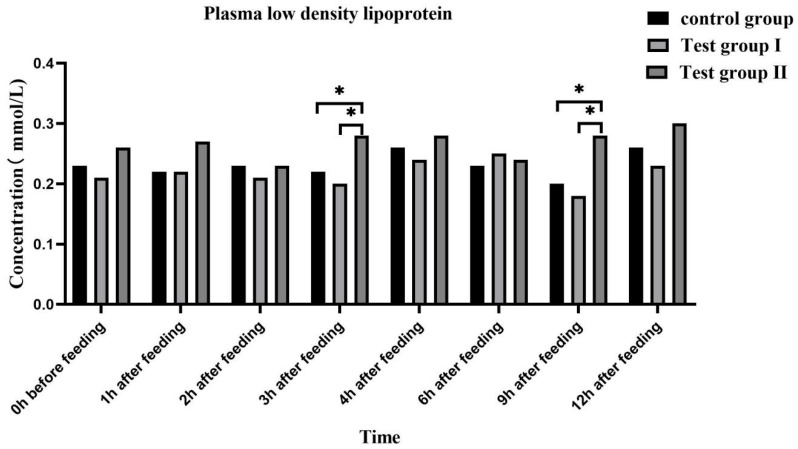
Effects of feed concentrate and FA supplementation on plasma LDL content in grazing *Yili* horses. * Significant differences.

**Figure 7 animals-13-02415-f007:**
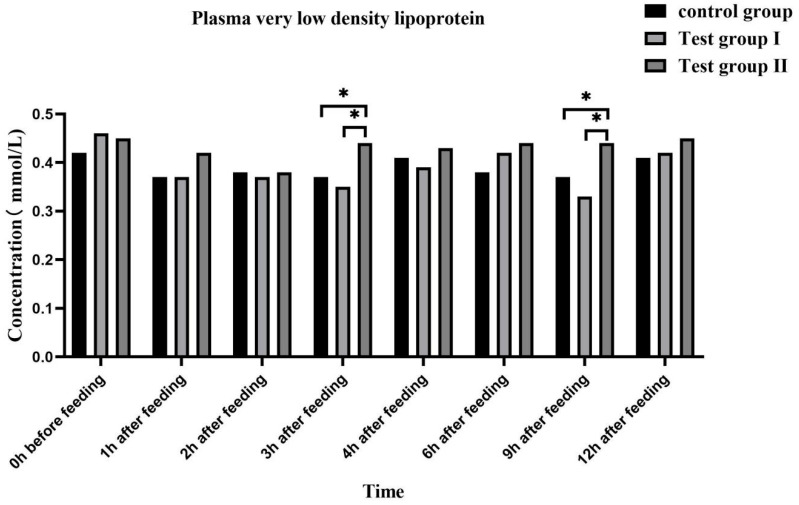
Effects of feed concentrate and FA supplementation on plasma VLDL content in grazing *Yili* horses. * Significant differences.

**Figure 8 animals-13-02415-f008:**
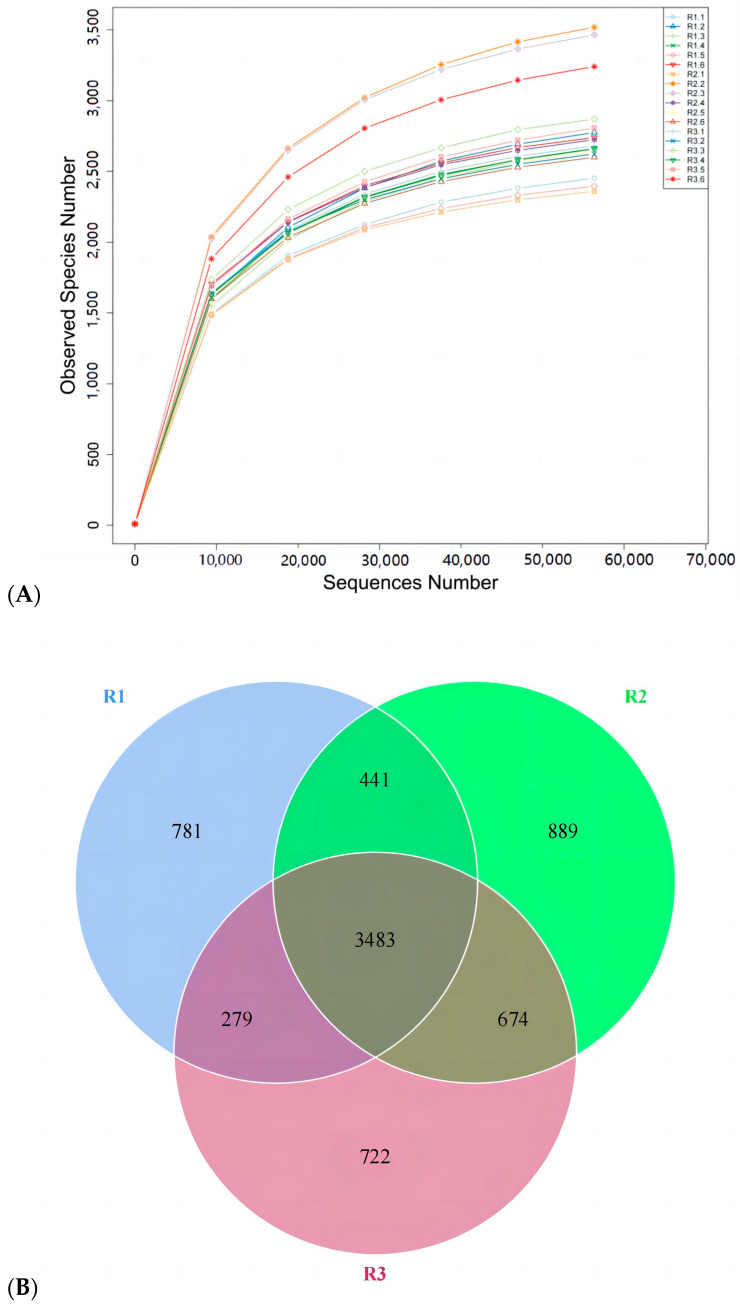
Dilution curve of intestinal microflora (**A**). Venn diagram of intestinal microflora (**B**).

**Figure 9 animals-13-02415-f009:**
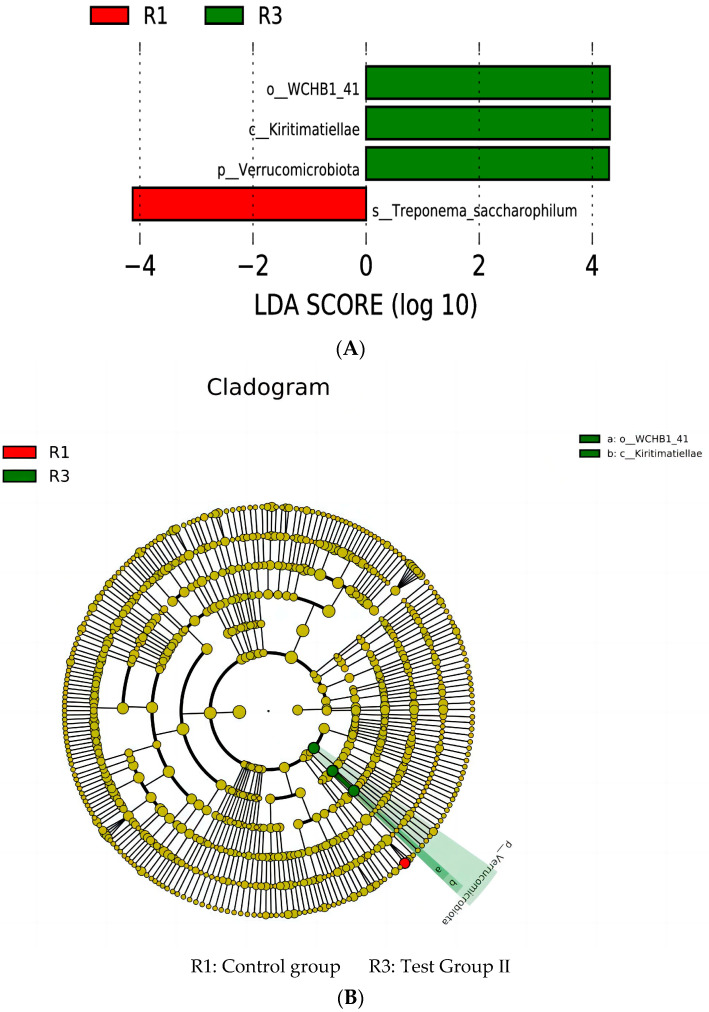
Histogram of linear discriminant analysis (LDA) value distribution (**A**). Evolutionary branch diagram (**B**).

**Figure 10 animals-13-02415-f010:**
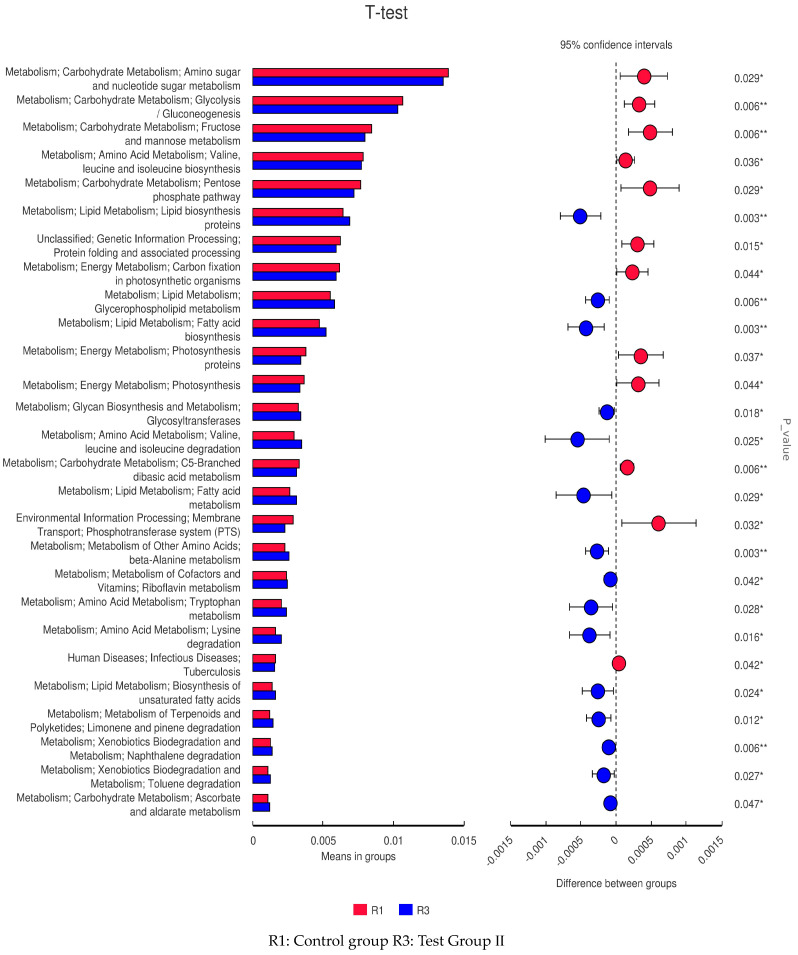
Level 3 functional prediction chart of differences between R1 and R3 according to PICRUSt *t*-test. * Significant differences; ** Very significant differences.

**Figure 11 animals-13-02415-f011:**
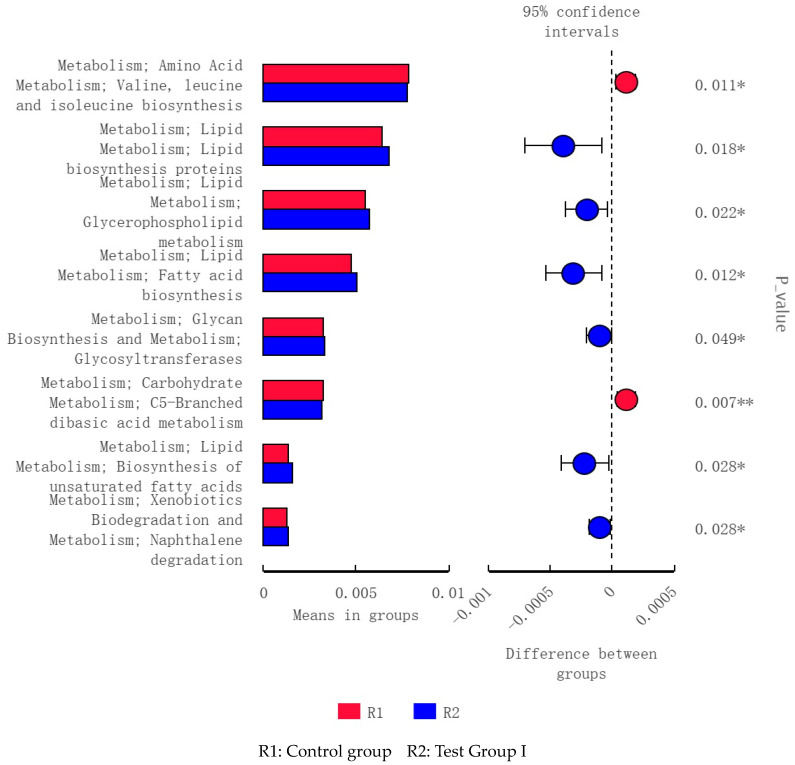
Level 3 functional prediction chart of differences between R1 and R2 according to PICRUSt *t*-test. * Significant differences; ** Very significant differences.

**Figure 12 animals-13-02415-f012:**
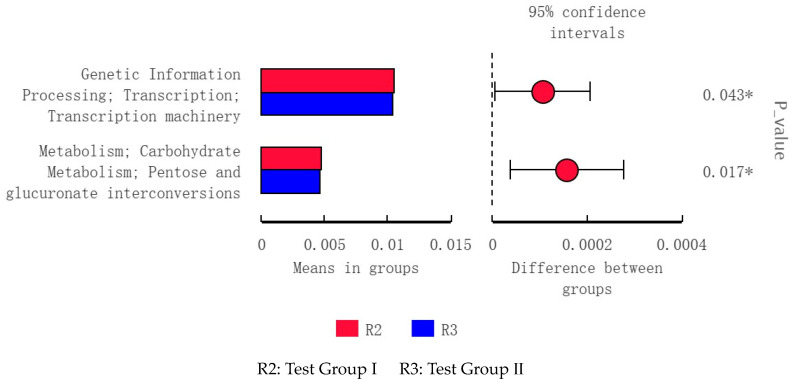
Level 3 functional prediction chart of differences between R2 and R3 according to PICRUSt *t*-test. * Significant differences.

**Table 1 animals-13-02415-t001:** Composition and nutrition level of concentrated feed supplement (dry matter basis) in terms of %.

Ingredient	Content	Nutrient ^(2)^	Content
Barley	55.54	Dry matter	89.25
Corn	36.00	Crude protein	14.56
Soybean meal	6.00	Ether extract	4.08
CaHPO_4_	1.30	Organic matter	97.18
Premix ^(1)^	1.16	Neutral detergent fiber	13.36
Total	100.00	Acid detergent fiber	5.18
--	--	Ash	2.82
--	--	Calcium	0.31
--	--	Phosphorus	0.46

^(1)^ The premix provided the following nutrients (per kilogram of concentrate): vitamin A, 14 mg; vitamin B1, 22 mg; vitamin B2, 332.1 mg; vitamin B6, 1.5 mg; vitamin D, 2.5 mg; vitamin E, 700 mg; biotin, 6 mg; pantothenic acid, 4.7 mg; nicotinamide, 12 mg; Cu, 42 mg; Fe, 113 mg; Mn, 186 mg; Zn, 176 mg; I, 28 mg; Se, 42 mg; Co, 4 mg. ^(2)^ Nutrient levels were measured.

**Table 2 animals-13-02415-t002:** Fatty acid composition of concentrated feed supplement, forage, and coated fatty acids in terms of %.

Items	Concentrate	Forage	Coated Fatty Acids
Butyric acid C4:0	0.43	0.82	0.24
Hexanoic acid C6:0	0.06	0.43	0.49
Octoic acid C8:0	0.25	1.17	0.37
Decanoic acid C10:0	0.00	0.00	5.25
Ricinoleic acid C11:0	0.00	0.00	0.02
Lauric acid C12:0	0.00	4.06	44.83
Tetradecanoic acid C14:0	0.12	0.86	18.42
Pentadecanoic acid C15:0	0.03	0.00	0.02
Palmitic acid C16:0	13.66	13.61	12.20
Palmitoleic acid C16:1	0.12	0.33	0.02
Margaric acid C17:0	0.05	0.32	0.01
Heptadecanoic acid monoenoic acid C17:1	0.03	0.30	0.00
Stearic acid C18:0	1.54	1.97	3.97
Elaidic acid C18:1n9t	0.03	0.00	0.00
Oleic acid C18:1n9c	25.17	3.02	3.91
Linoleic acid C18:2n6c	45.40	13.60	2.20
γ- Linolenic acid C18:3n6	0.45	4.07	0.05
α- Linolenic acid C18:3n3	2.44	28.67	0.56
Arachidic acid C20:0	0.39	2.66	0.12
Eicosaenoic acid C20:1	0.56	0.00	0.04
Cis-11,14-eicosadienoic acid C20:2	0.06	0.00	0.00
Behenic acid C22:0	0.20	1.11	0.00
Total saturated fatty acids ∑SFA	16.74	27	85.94
Total unsaturated fat acids ∑UFA	74.27	49.99	6.78
Total monounsaturated fatty acids ∑MUFA	25.92	3.65	3.97
Total unsaturated fat acids ∑PUFA	48.34	46.34	2.80
Total saturated fatty acid/total unsaturated fat acid ∑SFA/∑UFA	0.23	0.54	12.68

The measured value was the content of fatty acids.

**Table 3 animals-13-02415-t003:** Effects of feed concentrate and FA supplementation on the composition of grazing *Yili* horse milk (*n* = 6).

Items	Control Group	Test Group I	Test Group Ⅱ	SEM	*p*-Value		
					Grous	Time	G × T
Butterfat content (%)	1.47	1.45	1.49	0.07	0.944	0.104	0.897
Milk fat production (g/d)	3.59 ^B^	3.85 ^B^	4.77 ^A^	0.19	<0.001	0.088	0.793
Milk protein percentage (%)	1.63	1.61	1.58	0.04	0.643	0.918	0.982
Milk protein yield (g/d)	3.97 ^B^	4.27 ^B^	5.06 ^A^	0.11	<0.001	0.905	0.981
Lactose percentage (%)	6.68	6.76	6.76	0.04	0.307	0.563	0.555
Lactose production (g/d)	16.29 ^C^	17.90 ^B^	21.70 ^A^	0.11	<0.001	0.625	0.608
Total solids (%)	9.86	9.90	9.88	0.10	0.963	0.67	0.469
Somatic cell number (Thousand/mL)	21.25	13.33	18.25	4.67	0.489	0.331	0.418
Solid no fat (%)	8.54	8.58	8.57	0.06	0.858	0.654	0.681
Urea nitrogen (mg/dL)	26.64 ^aA^	24.24 ^bB^	24.58 ^bAB^	0.61	0.02	<0.001	0.967

Values with no letter or the same superscript letter within the same row do not significantly differ (*p* > 0.05). Values with different lowercase superscript letters within the same row significantly differ (*p* < 0.05). Values with different uppercase superscript letters within the same row very significantly differ (*p* < 0.01).

**Table 4 animals-13-02415-t004:** Effects of feed concentrate and FA supplementation on the % FA composition of grazing *Yili* horse milk.

Items	Control Group	Test Group I	Test Group Ⅱ	SEM	*p*-Value
Butyric acid C4:0	0.34	0.42	0.37	0.02	0.390
Hexanoic acid C6:0	0.22	0.20	0.21	0.01	0.895
Octoic acid C8:0	1.22	1.20	1.29	0.05	0.742
Decanoic acid C10:0	2.83	2.94	3.26	0.13	0.380
Ricinoleic acid C11:0	0.35	0.37	0.38	0.02	0.805
Lauric acid C12:0	3.93 ^b^	4.32 ^b^	8.78 ^a^	0.55	<0.001
Tetradecanoic acid C14:0	5.13 ^b^	5.64 ^b^	7.50 ^a^	0.27	<0.001
Myristoleic acid C14:1	0.77	0.77	0.91	0.03	0.091
Pentadecanoic acid C15:0	0.32 ^a^	0.29 ^ab^	0.23 ^b^	0.01	0.013
Palmitic acid C16:0	19.14	20.53	19.16	0.32	0.124
Palmitoleic acid C16:1	7.98	7.77	6.93	0.23	0.139
Margaric acid C17:0	0.18	0.17	0.16	0.01	0.738
Heptadecanoic acid monoenoic acid C17:1	0.61 ^a^	0.59 ^a^	0.46 ^b^	0.02	0.005
Stearic acid C18:0	0.67	0.73	0.78	0.03	0.265
Elaidic acid C18:1n9t	0.19	0.21	0.17	0.01	0.268
Oleic acid C18:1n9c	12.66	12.63	13.31	0.20	0.304
Linoleic acid C18:2n6c	7.90 ^a^	7.73 ^ab^	7.24 ^b^	0.12	0.056
γ- Linolenic acid C18:3n6	8.31 ^a^	7.90 ^a^	6.32 ^b^	0.26	<0.001
α- Linolenic acid C18:3n3	20.55 ^a^	18.74 ^a^	15.32 ^b^	0.62	<0.001
Cis-11,14-eicosadienoic acid C20:2	0.15 ^b^	0.13 ^b^	0.18 ^a^	0.01	0.002
Cis-8,11,14-eicosotrienic acid C20:3n6	0.16	0.15	0.17	0.01	0.704
Cis-11,14,17-eicosotrienic acid C20:3n3	0.41	0.40	0.39	0.01	0.928
Total saturated fatty acids ∑SFA	34.35 ^b^	36.81 ^b^	42.12 ^a^	0.94	<0.001
Total unsaturated fat acids ∑UFA	59.70 ^a^	57.03 ^a^	51.38 ^b^	1.04	<0.001
Total monounsaturated fatty acids ∑MUFA	22.22	21.97	21.77	0.28	0.824
Total unsaturated fat acids ∑PUFA	37.48 ^a^	35.06 ^a^	29.61 ^b^	0.93	<0.001
Total saturated fatty acid/total unsaturated fat acid ∑SFA/∑UFA	0.58 ^b^	0.65 ^b^	0.82 ^a^	0.03	<0.001

Values with no letter or the same superscript letter within the same row do not significantly differ (*p* > 0.05). Values with different lowercase superscript letters within the same row significantly differ (*p* < 0.05).

**Table 5 animals-13-02415-t005:** Analysis of alpha diversity of fecal bacteria in grazing *Yili* horses supplemented with feed concentrate and FAs.

Items	Control Group	Test Group I	Test Group Ⅱ	SEM	*p*-Value
Observed species	2163.67	2884.33	2814.00	76.73	0.348
Shannon index	9.30	9.59	9.53	0.06	0.108
Simpson index	0.99	1.00	1.00	0.0006	0.426
Chao1 index	2811.71	3084.11	3026.10	81.91	0.382
ACE index	2829.28	3107.67	3045.80	83.87	0.386
Goods coverage (%)	0.99	0.99	0.99	0.0003	0.883

**Table 6 animals-13-02415-t006:** Effects of feed concentrate and FA supplementation on the horizontal fecal microflora abundance % in grazing *Yili* horses.

Items	Control Group	Test Group I	Test Group Ⅱ	SEM	*p*-Value
Bacteroidetes	44.20	42.75	43.51	1.30	0.911
Firmicutes	32.59	33.60	31.52	1.02	0.736
Spirochaetes	6.41	3.39	2.98	0.87	0.221
Verrucomicrobia	3.92 ^c^	6.81 ^b^	7.92 ^a^	0.63	0.017
Unidentified_Bacterri	2.27	2.61	3.01	0.21	0.372
Proteobacteria	1.20	1.66	2.17	0.25	0.309
Euryarchaeota	0.66	0.43	0.52	0.14	0.817
Halobacterota	0.72	0.43	0.52	0.14	0.248
Fibrobacterota	1.25	0.87	1.00	0.09	0.186
Acidobacteriota	0.18	0.32	0.23	0.08	0.804
Others	6.60	7.14	7.05	0.35	0.817

Values with no letter or the same superscript letter within the same row do not significantly differ (*p* > 0.05). Values with different lowercase superscript letters within the same row significantly differ (*p* < 0.05).

**Table 7 animals-13-02415-t007:** Effects of feed concentrate and FA supplementation on the fecal bacterial family abundance % in grazing *Yili* horses.

Items	Control Group	Test Group I	Test Group Ⅱ	SEM	*p*-Value
Rikenellaceae	11.95	12.84	13.77	0.98	0.773
Lachnospiraceae	11.18	10.39	9.96	0.61	0.733
Spirochaetaceae	6.26	3.25	2.82	0.86	0.213
Prevotellaceae	8.39	6.42	5.88	0.54	0.138
p-251-o5	8.72	6.34	6.44	0.67	0.279
F082	6.04	7.80	7.62	0.39	0.123
Bacteroidales_RF16_group	2.37	2.30	2.95	0.29	0.632
Oscillospiraceae	3.55	4.36	4.25	0.17	0.107
Clostridiaceae	0.96	0.48	0.52	0.22	0.644
Ruminococcaceae	2.42	2.43	2.46	0.13	0.992
Others	38.16	43.38	43.32	1.16	0.104

**Table 8 animals-13-02415-t008:** Effects of feed concentrate and FA supplementation on the fecal bacterial genus abundance % in grazing *Yili* horses.

Items	Control Group	Test Group I	Test Group Ⅱ	SEM	*p*-Value
Treponema	6.09	3.13	2.74	0.85	0.219
Rikenellaceae_RC9_gut_group	9.33	9.60	10.33	0.69	0.849
Clostridium_sensu_stricto_1	0.90	0.34	0.34	0.22	0.518
Prevotellaceae_UCG-001	1.78	1.11	1.02	0.18	0.171
UCG-004	0.89	0.63	0.65	0.15	0.750
Prevotellaceae_UCG-004	1.24	1.61	1.49	0.11	0.405
Ruminococcus	2.00	1.87	1.80	0.12	0.810
Prevotellaceae_UCG-003	1.85	1.21	1.21	0.13	0.065
Lachnospiraceae_UCG-009	1.21	0.89	0.71	0.12	0.230
Faecalibaculum	0.53	0.44	0.43	0.17	0.227
Others	73.98 ^b^	79.17 ^a^	79.72 ^a^	1.00	0.024

Values with no letter or the same superscript letter within the same row do not significantly differ (*p* > 0.05). Values with different lowercase superscript letters within the same row significantly differ (*p* < 0.05).

## Data Availability

The data that support the findings of this study are available from the corresponding author upon reasonable request.

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
