# Peer review of "Effect of Supplementary Feeding on Milk Volume, Milk Composition, Blood Biochemical Index, and Fecal Microflora Diversity in Grazing Yili Mares"

_animals, 2023, doi:10.3390/ani13152415_

Round 1

Reviewer 1 Report

No comments

Very poor english

Reviewer 2 Report

REVIEW

“Effect of supplementary feeding on milk volume, milk composition, blood routine, and fecal flora diversity of grazing Yili horses”

BRIEF SUMMARY AND GENERAL COMMENTS:

The present study describes the effects of two different types of feeding supplementation on Yili mares’ lactation and fecal microbiota. The study is potentially interesting, as there are few studies in the literature on milk horses, and it was conducted on a specific Chinese breed used for milk production. The study design was appropriate, and some interesting results were found. However, the manuscript is very poorly written. English language is very poor, and there are several grammar and syntaxis mistakes, that sometimes make the manuscript unreadable and hard to understand. Moreover, it was not possible for me to review some parts of the manuscript as I could not understand the meaning of the content due to poor English. Moreover, there are some sentences throughout the manuscript that cannot stand alone, as subjects and/or verbs are missing, suggesting that the authors have not even re-read the manuscript before submitting it. Major and extensive English editing is needed before considering the manuscript further.

SPECIFIC COMMENTS:

Simple summary

Overall, the simple summary is not clear. I suggest you should add a brief sentence on what kind of supplementary feeding was given to horses (the two types of supplementations used in different groups should be mentioned).

-        Lines 10-12: the first sentence must be rephrased. I suggest rephrasing like this: “In the present study, the effects of supplementary feeding on milk yield, milk composition, plasma biochemical indices, and fecal flora in a population of grazing Yili horses were evaluated.”

-        Lines 12-15: I suggest splitting the sentence after “supplementary feeding” and before “which had”. I.e. “… after supplementary feeding. In particular, supplementary feeding showed an effect on the improvement of the fatty acids in horses’ milk and on the increase of glucose plasma concentration.”

Abstract

Abstract is too long: Animals requires an abstract of maximum 200 words. It should be completely re-written before I can consider it for review.

Introduction

Overall, the content of the introduction is clear, but English quality is poor and it is not well written. Please, find below some point-by-point comments.

-        Line 57: The verb is missing: “Dairy Yili horses is a breed that is independently bred by…”

-        Line 59: I suggest replacing “…can be in harsh…” with “…can live in harsh…”

-        Line 60: Yield and quality are plural; plural verbs should be used. “The yield and quality of horse milk are important factors not only for determining…” and “…also have…”

-        Line 62: I understand your point, but I suggest you should explicitly write that horse milk can represent a valid option for human nutrition too, because it has similar composition to human milk and it is easy to absorb for people.

-        Lines 63-65: So, the problem is that the produced milk is not enough for the growth of foals, or for the human use? Please clarify.

-        Lines 69-71: Please rephrase as follows: “The improvement of the lactation amount and the quality of milk components by nutritional supplementation during the lactation period are important to promote the performance of mares and the healthy growth of foals”

-        Line 84:  Please rephrase as follows: “…which has an important role in the maintenance of the intestinal…”

-        Lines 86-87: where is the verb? Rephrase: “At present, research on the effects of supplementary feeding on milk production performance of Yili horses is lacking”

-        Lines 87-91: All this sentence must be rephrased, as it makes no sense in the present form. I suggest something like: “Therefore, in the present study, the effects of supplementary feeding on the lactation yield, milk composition and fecal flora diversity of grazing Yili horses were investigated, in order to provide a reference for improving the milk yield and quality, and promoting the healthy growth and development of foals.”

Materials and methods

In general, some parts of the materials and methods are not clear at all. There are major issues with the English language, which make it very difficult for the reader to understand how the study was conducted.

-        Line 97: If a sentence starts with a number, it should be spelled out, i.e. “Eighteen healthy Yili mares with similar…”

-        Line 99: “…with each group including six mares.”

-        Lines 100-103: It is not clear what the test groups were fed. What do you mean with “on the basis of the control group”? I think you are meaning that the horses in Group I were fed as the control group plus 1 kg/die of concentrate, and that horses in Group II were fed as horses in Group I plus 0.4 kg/die coated fatty acids. Is it correct? Please rephrase this sentence, as it is not clear at all.

-        Lines 105-112: Everything is very confusing in this paragraph. Did the horses wear a pocket containing food??? It is not clear. And what does this sentence mean? “After the horses in groups I and II fed concentrate, all the horses picked the pocket.”. Also, please state that milk collection started at 11:00 am every day and that milk was collected every 2 hours until 7:00 pm. In the present form, it is not clear. Also, I do not understand the sentence “After 3 days of pre-feeding, begin the formal test, all horses were managed uniformly and consumed food and water freely”. The whole paragraph should be re-written.

-        Table 1: I think that Ingredient and Nutrient Levels should be in the first row of the table instead of “items”. Moreover, in the footing of the table, the meaning of the acronyms used in the Table should be spelled out (DM, CP, EE, OM, etc.)

-        Line 125: Were the milk samples collected every 7 days at a specific hour, or did it change between different collections?

-        Line 129: How could the stomach be empty if horses live at pasture?

-        Line 131: Please replace 4000 rpm with g, as rpm is not a valid measure (it depends on the diameter of you centrifuge).

-        Lines 134-137: Please rephrase “On day 45 and 46 of the study, feces were collected from horses at 11:30 am and 7:30 pm. After 2-day collection, feces were mixed, 10% was randomly selected, weghed, and air-dried”. What do you mean with “crushed in 0.63 mm”? It is not clear. Then, the last sentence should be rephrased as follows: “Moreover, 5 g of fresh feces samples were stored in 5 mL frozen storage tubed with liquid nitrogen for testing”.

-        Lines 140-150: These two paragraphs are not clear at all. Please re-write them.

-        Line 152: Please correct: “On day 40 of the study, the plasma…”

-        Line 158: Be consistent with the verb tenses. “Draws” is present while in the rest of the manuscript you used the past.

-        Lines 160-162: What does the part in brackets mean?

-        Lines 165-169: The meaning is clear, but it should be re-written. You cannot use commas for everything, and then start with capital letters after commas. Separate the sentences with dots.

Results

In general, some results are not clearly presented (see specific comments below). In the titles of the subheadings, sometimes you use “effect” and sometimes “effects”: please, choose one form and be consistent. Moreover, throughout the results section,           you report only that p value was < 0.05, but it should be nice to know the exact p values provided by the statistical softwares used.

-        Line 175: remove “were” before “peaked”.

-        Lines 176-178: Why do you mention the two figures separately? Indeed, they both show the differences between the 3 groups, one on the average milk production and the other on the total milk production. You should state it in the main text.

-        Line 187: What is shown in Table 3? You should write it in the main text. I.e. “The effects of concentrate and fatty acid supplementations on milk composition of grazing Yili horses are shown in Table 3.”

-        Line 188: Before “The milk fat, milk protein,…” you should put a dot and separate sentences.

-        Lines 188-192: It is confusing how you reported the results in this sentence. You say it was higher than controls and test group I, but you do not mention test group II. Please, pay attention to the subjects and verbs used throughout the manuscript, as the syntaxis of your sentences if very often confusing.

-        Line 197: Remove “The same as below”. You should report the same footnotes (about meaning of superscript letters) also in the following Table.

-        Line 200: As above, mention in the main text what is included in Table 4.

-        Lines 200-205: As above, you should divide sentences to make it clearer for the reader. Moreover, you should pay attention to the correct use of capital letters and punctuation.

-        Lines 210-213: As above, divide sentences.

-        Line 256, 266, 277, 285: As above, mention in the main text what is included in Table 5.

-        Line 270: It is not clear in which group Verrucomicrobia was higher than the control group?

-        Line 277 and 285: “In this experiment, the top 10 family level was detected in the family level” and “the top 10 genus level was detected in the family level”: what does it mean?

-        Line 289: Same as above, in which group Prevotellaceae were lower than the control group?

-        Lines 304-308: As above, divide sentences with dots and use capital letters properly.

Discussion

-        Line 326: please correct “grazing Conditions. Energy intake…”

-        Line 329: “daily milk viewer”???

-        Line 332-333: “In the last three weeks of the experiment.” This sentence cannot stand alone. What do you mean?

-        Lines 333-335: Please rephrase as follows: “The milk production of mares in the group I was significantly different from that of the control group; the reasons for this finding may be due to lack of rainfall, sparse grass quantity, grass quality is poor, and the late lactation of the mares.”

-        Lines 338-339: “In this experiment, no significant changes between groups of the milk composition.” Where is the verb?

-        Line 340-341: “The reason maybe that priority participation of fatty acids in hydrolysis to promote lactation process.” I understand this sentence but is not correct in English. Please rephrase it.

-        Lines 348-351: “In this experiment, the contents of lauric acid and myristic acid in experimental group II were significantly improved, to a certain extent, it can improve the immune system and digestive metabolism of mares, thus can improve horse milk production” As above, I understand but this sentence need to be re-written. Words are missing to link the two parts of the sentence.

-        Lines 354-360: Once again, these sentences need to be rephrased.

-        … and many more. All discussion needs to be extensively re-phrased to make it understandable for the reader. The content is fine, but English is too poor.

English quality is very poor and even difficult to understand. 

Author Response

Dear reviewer,

We have revised and polished the article

Thanks a lot!

Reviewer 3 Report

Dear authors,

I have reviewed your study and would like to provide a critical comment on the content. While your study explores the effects of supplementary feeding on various parameters in Yili horses under grazing conditions, the presentation of the results and their interpretation could be improved for better clarity and comprehension.

To enhance the readability and understanding of your findings, I suggest organizing the results section in a more structured manner. Presenting the results in a clear and concise format, such as using tables or figures, would facilitate the readers' comprehension and enable them to extract key information easily.

Furthermore, it would be beneficial to provide a more detailed discussion of the implications and significance of your findings. For example, you mention that the lactation volume of Yili horses in the test groups was significantly higher than that of the control group, but it would be valuable to discuss the practical implications of this increase in milk production and its potential benefits for horse breeders or the industry.

Additionally, when discussing the changes in blood biochemical indexes, milk components, and fecal microflora diversity, it would be helpful to provide a more comprehensive analysis of the observed differences between the groups. Exploring the potential mechanisms behind these changes and relating them to previous research or established knowledge in the field would enhance the scientific rigor of your study.

Overall, by reorganizing and expanding upon the presentation and discussion of your results, you can improve the clarity and impact of your research. This will enable readers to better understand the implications of your findings and their relevance to the field of Yili horse breeding.

Specific comments:

Upon reviewing your paper, I would like to provide feedback on the simple summary you have included. It appears that the current simple summary does not align with the guidelines set by the journal.

To meet the journal's guidelines, a simple summary should be concise and accessible to readers who may not be familiar with the research field. It should provide a clear overview of the study's objectives, methods, key findings, and their significance. The current simple summary lacks the necessary brevity and clarity required for a concise summary.

I recommend revising the simple summary to adhere to the journal's guidelines. Ensure that it effectively communicates the main aspects of your study in a succinct and understandable manner. This will help readers grasp the essence of your research and its implications.

I would like to provide feedback on the length of the abstract. It appears that the abstract is quite lengthy and may exceed the recommended length for abstracts in the journal.

A well-written abstract should provide a concise summary of the study's objectives, methods, key findings, and conclusions. It is meant to provide a brief overview of the entire paper, giving readers a clear idea of the research without delving into excessive detail.

I recommend revising the abstract to ensure it is more concise and focused. Identify the most important points and findings from your study and present them in a clear and succinct manner. Avoid including unnecessary background information or excessive technical details that can be better addressed in the main body of the paper.

By condensing the abstract, you will make it more accessible to readers and increase its effectiveness in conveying the main contributions of your research.

Could you please provide further clarification or expand on the concept that horse milk is easily absorbed? It would be helpful to include specific details or evidence supporting this claim in your paper. Additionally, if there are any comparative studies or physiological mechanisms that explain the superior absorbability of horse milk, including that information would enhance the understanding of your findings.

I appreciate the inclusion of the concept of palatability in your paper. However, it would be beneficial to expand on this concept and provide more details regarding its significance in horse feed. Specifically, you could discuss the factors that contribute to palatability, such as taste, texture, and aroma, and how they influence the acceptance and consumption of horse feed.

To support your claims about the importance of palatability, I suggest referencing a specific study that has explored this topic in the context of horse feeding. For example, you could refer to the following reference: 10.1016/j.applanim.2020.105110.

This reference will provide readers with a more in-depth understanding of the role of palatability in horse feed and support your statements in a scientifically sound manner.

Including this additional information will enhance the quality and comprehensiveness of your paper. Thank you for considering this suggestion.

I agree that it is important to address the potential negative effects of excessive concentrate feeding on horse health and the increased risk of certain pathologies. It would be valuable to include more emphasis on this aspect in your paper.

To strengthen your discussion on the negative effects of excessive concentrates, I recommend citing the following references that provide valuable information on this topic:

10.3390/ani12141740 and 10.1186/s12917-022-03289-2

These references will provide readers with further insights into the potential health risks associated with excessive concentrate feeding in horses and support your arguments. Specifically, 10.3390/ani13061107 mentioned regarding the increased risk of colics would be relevant in highlighting the potential consequences of imbalanced concentrate feeding.

By incorporating these references, you can provide a more comprehensive analysis of the potential drawbacks of excessive concentrate feeding and contribute to a better understanding of the topic.

the ethical approvement is missing

the quality of the pasture plays a crucial role in the overall findings of your study. It is important to provide a detailed description of the pasture and address the presence and evaluation of unwanted weeds.

To enhance the completeness of your paper, I recommend citing the following valuable paper that specifically addresses these aspects:

10.1016/j.jevs.2022.103940

This paper will provide valuable insights into the evaluation of pasture quality and the presence of unwanted weeds, allowing readers to better understand the environmental conditions under which your study was conducted.

By incorporating this reference, you can improve the clarity and comprehensiveness of your paper and provide a more comprehensive picture of the pasture conditions and potential challenges associated with unwanted weeds.

I have a question regarding the analysis of aflatoxin levels in the concentrates used in your study.

Aflatoxin contamination in animal feed can have detrimental effects on animal health. It would be beneficial if you could provide information on whether aflatoxin levels were assessed in the concentrates you used and, if so, at what levels. Evaluating aflatoxin content in concentrates is crucial to understanding the potential risks associated with its consumption.

Additionally, I suggest considering the following paper, which provides valuable insights into aflatoxin contamination in animal feed:

10.3390/toxins14070430

This paper discusses the significance of aflatoxin contamination in animal feed and its impact on animal health. By incorporating this reference into your study, you can enhance the discussion on the quality and safety of the concentrates used, particularly in relation to aflatoxin levels.

I kindly ask you to review whether aflatoxin analysis was performed in your study and, if so, provide details regarding the levels detected. Including this information will strengthen your study's findings and contribute to the understanding of potential risks associated with aflatoxin contamination.

I would like to request further clarification regarding the analysis of milk, blood, and microbiota samples in your study.

  1. Milk Analysis: Could you please provide more details on the specific parameters that were analyzed in the milk samples? This would help us understand the comprehensive evaluation of milk quality and composition resulting from the different feeding regimens.

  2. Blood Biochemical Indexes: It would be helpful if you could elaborate on the specific blood biochemical indexes that were measured in the study. Additionally, please provide information on the rationale behind selecting these particular indexes and their relevance to assessing the effects of supplementary feeding on horse health and performance.

  3. Fecal Microbiota Diversity: In your study, you investigated the fecal microflora diversity of Yili horses. Could you please provide more specific details on the methods used for analyzing the fecal microbiota, such as the sequencing technique employed and the specific taxonomic groups that were identified? Additionally, any relevant findings or patterns observed in the fecal microbiota diversity would greatly contribute to the understanding of the impact of supplementary feeding on gut health.

By providing additional information and elaborating on these aspects, you can further enhance the scientific rigor and comprehensiveness of your study. This will allow readers to gain deeper insights into the effects of supplementary feeding on milk quality, blood parameters, and gut microbiota in Yili horses.

I have a specific question regarding the statistical analysis of your data.

I would like to inquire if you performed a normality test on the collected data. Assessing the normality of the data is crucial in determining the appropriate statistical tests and ensuring the validity of the analysis. If you conducted a normality test, could you please provide information on the specific test used and any relevant results? This information would greatly contribute to the transparency and reliability of your study.

If a normality test was conducted, I suggest citing an appropriate reference, such as the following:

  • Reference: 10.1080/1828051X.2020.1827990

Including this information will enhance the methodological robustness of your study and allow readers to better understand the statistical analysis employed.

I have a question regarding the behavior of the horses during milking and their overall management.

Could you please provide more details about the behavior of the horses during the milking process? It would be valuable to know if any specific behaviors or reactions were observed, such as calmness, restlessness, or resistance during milking. Additionally, if any specific management techniques were employed to ensure the well-being and cooperation of the horses during the milking process, it would be beneficial to include that information as well.

Enhancing our understanding of the horses' behavior and management during milking will provide further insights into the practical aspects of your study and contribute to the overall interpretation of the results.

I would like to kindly request you to include a section discussing the practical implications and limitations of your study.

Including a practical implications section would help readers understand the real-world applications of your research findings. It would be beneficial to discuss how the results of your study can be applied in horse husbandry practices, such as optimizing supplementary feeding strategies, improving milk production, or enhancing the overall health and well-being of Yili horses.

Furthermore, addressing the limitations of the study would provide a more comprehensive perspective to the readers. It is important to acknowledge any constraints, potential biases, or challenges encountered during the research process. This would help in interpreting the results and provide insights for future studies or modifications in methodology.

I believe that including these sections will enhance the practical relevance of your study and provide a more well-rounded understanding of its implications.

Thank you for considering this suggestion. I appreciate your efforts in conducting this research.

Author Response

(The authors gave the same response as above.)

Round 2

Reviewer 1 Report

In my opinion the paper, after the revisions made by the authors, seems to be much improved and ready for publication in the present form

Author Response

Dear reviewer

Thanks a lot

Reviewer 2 Report

REVIEW - 2nd round

“Effect of supplementary feeding on milk volume, milk composition, blood routine, and fecal flora diversity of grazing Yili horses”

BRIEF SUMMARY AND GENERAL COMMENTS:

The quality of the manuscript has greatly improved, and English language is overall fine now. Some minor corrections are still needed before accepting it for publication, and I still have a major concern about how horses were fed during the experiment (see materials and methos, paragraph 2.3). It is not clear to me, preventing me from interpreting results accordingly.  

SPECIFIC COMMENTS:

Simple summary

The simple summary is clear now, but I suggest adding a brief sentence describing the kind of supplementary feeding given to horses.

Abstract

The abstract is good, but a brief background sentence should be added before describing the study, to allow the reader understanding the reasons behind your study.

Introduction

The introduction is clear. Just be careful, at Line 52 “High density breeding of horses” (horses should be plural).

Materials and methods

-       The paragraph 2.3 is still not clear to me. During the 3-day pre-feeding, horses received their pocket that could be empty, or full of concentrate or concentrate + fat powder, based on the group. Later, when the experiment started, all horses were given ad libitum. So, I don’t understand how could you evaluate the effects of different supplementation, if horses were managed all in the same way after the first 3 days? The experiment design needs to be explained better.

-       Lines 104-108: Please rephrase as follows: “When milking, the mares were taken to the enclosure of the foals, the foals were led to the mares and let suck the nipples for 2-5 seconds to stimulate the mammary gland to secrete milk. They, foals and mares were separated, and the milking staff began to collect the mares’ milk, keeping the foals beside the mares to calm them, with the aim to collect more horse’s milk.”

-       Line 113: 4000 rpm should be converted to g.

-       Line 118: Please correct: “Five grams of fresh feces were placed…”

-       Line 133: Please correct: “On day 40, plasma was collected at 0, 1, 2, 3, 4, 6, 9, and 12 h after the morning feeding, and sent to…”

Results

Could you please report in the main text exact p values, and not only “p < 0.05”?

Discussion

-       Lines 341-344: this sentence has syntaxis problems, starting from “…the ash yeld, the total solid and lactose…”. Please rephrase it.

-       Lines 345, 352, 356: there are some punctuation problems to be addressed.

-       Lines 432-433: I don’t understand what do you mean with this sentence? “However, mares could not guarantee a calm state under grazing conditions during the course of this experiment.”?

English language is fine, except from some minor mistakes.

Author Response

BRIEF SUMMARY AND GENERAL COMMENTS:

The quality of the manuscript has greatly improved, and English language is overall fine now. Some minor corrections are still needed before accepting it for publication, and I still have a major concern about how horses were fed during the experiment (see materials and methos, paragraph 2.3). It is not clear to me, preventing me from interpreting results accordingly.  

SPECIFIC COMMENTS:

Simple summary

The simple summary is clear now, but I suggest adding a brief sentence describing the kind of supplementary feeding given to horses.

supplementary feeding is beneficial for grazing mares

Abstract

The abstract is good, but a brief background sentence should be added before describing the study, to allow the reader understanding the reasons behind your study.

Grazing is a common rearing pattern

Introduction

The introduction is clear. Just be careful, at Line 52 “High density breeding of horses” (horses should be plural).

Has been modified

Materials and methods

-       The paragraph 2.3 is still not clear to me. During the 3-day pre-feeding, horses received their pocket that could be empty, or full of concentrate or concentrate + fat powder, based on the group. Later, when the experiment started, all horses were given ad libitum. So, I don’t understand how could you evaluate the effects of different supplementation, if horses were managed all in the same way after the first 3 days? The experiment design needs to be explained better.

After the horses are grouped, the horses were prefed for 3 days

-       Lines 104-108: Please rephrase as follows: “When milking, the mares were taken to the enclosure of the foals, the foals were led to the mares and let suck the nipples for 2-5 seconds to stimulate the mammary gland to secrete milk. They, foals and mares were separated, and the milking staff began to collect the mares’ milk, keeping the foals beside the mares to calm them, with the aim to collect more horse’s milk.”

Has been modified

-       Line 113: 4000 rpm should be converted to g.

Has been modified

-       Line 118: Please correct: “Five grams of fresh feces were placed…”

-       Line 133: Please correct: “On day 40, plasma was collected at 0, 1, 2, 3, 4, 6, 9, and 12 h after the morning feeding, and sent to…”

Has been modified

Results

Could you please report in the main text exact p values, and not only “p < 0.05”?

I didn't understand the question

Discussion

-       Lines 341-344: this sentence has syntaxis problems, starting from “…the ash yeld, the total solid and lactose…”. Please rephrase it.

Has been modified

-       Lines 345, 352, 356: there are some punctuation problems to be addressed.

Has been modified

-       Lines 432-433: I don’t understand what do you mean with this sentence? “However, mares could not guarantee a calm state under grazing conditions during the course of this experiment.”?

However, the mare could not be guaranteed in a calm state at each milking under grazing conditions during this experiment.

Reviewer 3 Report

good job

Author Response

Dear reviewer

Thanks a lot